# *NF2* Alteration/22q Loss Is Associated with Recurrence in WHO Grade 1 Sphenoid Wing Meningiomas

**DOI:** 10.3390/cancers14133183

**Published:** 2022-06-29

**Authors:** Yu Sakai, Satoru Miyawaki, Yu Teranishi, Atsushi Okano, Kenta Ohara, Hiroki Hongo, Daiichiro Ishigami, Daisuke Shimada, Jun Mitsui, Hirofumi Nakatomi, Nobuhito Saito

**Affiliations:** 1Departments of Neurosurgery, Faculty of Medicine, The University of Tokyo, Tokyo 113-8655, Japan; sakai-yu@umin.ac.jp (Y.S.); yteranishi-nsu@umin.ac.jp (Y.T.); okano-cib@umin.ac.jp (A.O.); ohara-tmd@umin.ac.jp (K.O.); hongou-sin@umin.ac.jp (H.H.); dishigami-tky@umin.ac.jp (D.I.); nsaito-nsu@m.u-tokyo.ac.jp (N.S.); 2Department of Neurosurgery, Faculty of Medicine, Kyorin University, Tokyo 113-8655, Japan; dai-8264@umin.ac.jp (D.S.); hirofumi-nakatomi@ks.kyorin-u.ac.jp (H.N.); 3Department of Molecular Neurology, Graduate School of Medicine, The University of Tokyo, Tokyo 113-8655, Japan; mituij-tky@umin.ac.jp

**Keywords:** sphenoid wing meningioma, recurrence, *NF2*

## Abstract

**Simple Summary:**

Sphenoid wing meningiomas account for 11–20% of all intracranial meningiomas and have a higher recurrence rate than those at other sites. The molecular background of meningiomas has been clarified in recent years; therefore, in this study, we aimed to analyze factors associated with the recurrence of sphenoid wing meningiomas. We found that *NF2* alteration/22q loss is associated with recurrence in WHO grade 1 sphenoid wing meningiomas, as well as some radiological findings. Together, our study suggests that genotyping meningiomas could help to inform treatment strategies and play an important role in precision medicine, while the preoperative prediction of genotype from radiological features could influence surgical strategies. Furthermore, appropriate follow-up planning could improve the appropriate distribution of medical resources and provide both medical and economic benefits.

**Abstract:**

Sphenoid wing meningiomas account for 11–20% of all intracranial meningiomas and have a higher recurrence rate than those at other sites. Recent molecular biological analyses of meningiomas have proposed new subgroups; however, the correlation between genetic background and recurrence in sphenoid wing meningiomas has not yet been fully elucidated. In this study, we evaluated the clinical characteristics, pathological diagnosis, and molecular background of 47 patients with sphenoid wing meningiomas. Variants of *NF2, AKT1, KLF4, SMO, POLR2A, PIK3CA, TRAF7,* and *TERT* were determined using Sanger sequencing, and 22q loss was detected using multiplex ligation-dependent probe amplification. Alterations were localized at *NF2* in 11 cases, had other genotypes in 17 cases, and were not detected in 12 cases. Interestingly, WHO grade 1 meningiomas with *NF2* alteration/22q loss (*p* = 0.008) and a MIB-1 labeling index > 4 (*p* = 0.03) were associated with a significantly shorter recurrence-free survival, and multivariate analysis revealed that *NF2* alteration/22q loss was associated with recurrence (hazard ratio, 13.1). The duration of recurrence was significantly shorter for meningiomas with *NF2* alteration/22q loss (*p* = 0.0007) even if gross-total resection was achieved. Together, these findings suggest that *NF2* alteration/22q loss is associated with recurrence in WHO grade 1 sphenoid wing meningiomas.

## 1. Introduction

Sphenoid wing meningiomas account for 11–20% of all intracranial meningiomas [1] and are adjacent to important neurovascular structures, including the internal carotid artery, anterior choroidal artery, cavernous sinus, optic nerve, and cranial oculomotor nerves. These structures can limit gross-total resection (GTR), which is a key determinant of recurrence [2,3,4]. Indeed, it has been reported that the extent of resection (EOR) is associated with the recurrence of sphenoid wing meningiomas [5], which have a higher rate of recurrence than meningiomas in other locations [6].

Molecular genetic studies have revealed that *NF2* gene variants and 22q loss occur in approximately 40–60% and 60–70% of sporadic meningiomas, respectively [7,8,9,10]. Recent studies have also reported that *TRAF7*, *KLF4*, *AKT1*, *SMO*, *PIK3CA*, and *POLR2A* variants are all mutually exclusive of *NF2* variants [11,12,13,14]. In addition, *TERT* promoter (*TERTp*) variants have been associated with a shorter time to progression in meningiomas [15] and have been included as an independent criterion for WHO grade 3 meningiomas in the 2021 WHO Classifications of Tumors of the Central Nervous System [16]. In the last decade, a series of reports have been published on meningiomas related to copy number variants (CNVs), methylomes, and transcriptomes, and new subgroups have been proposed based on several molecular biological perspectives [17,18,19,20,21,22]. However, the association between the genetic background of sphenoid wing meningiomas and their recurrence has not yet been fully elucidated. In this study, we aimed to analyze factors associated with the recurrence of sphenoid wing meningiomas.

## 2. Materials and Methods

### 2.1. Study Cohort

We retrospectively analyzed the data of patients who underwent sphenoid wing meningioma resection at the University of Tokyo Hospital between April 2002 and December 2020. Sphenoid wing meningiomas were defined as those with dural attachment of the lesser wing of the sphenoid bone [23]. Patients with neurofibromatosis type 2, multiple meningiomas, and craniofacial meningiomas were excluded. If the patient underwent multiple surgeries, only data from the first surgery were used.

This study included 47 patients with sphenoid wing meningiomas whose characteristics are listed in Table 1. The cohort included 15 men and 32 women, with a median age of 57 (49–64) years and a median follow-up month of 27 (12–61.5). Of the 47 patients, 43 had WHO grade 1 meningiomas, three had WHO grade 2 meningiomas, one had a WHO grade 3 meningioma, and none had en plaque meningiomas. Although the grade 3 tumor was histologically classified as a meningothelial meningioma, it had a *TERTp* variant and was therefore classed as grade 3 according to the 2021 WHO Classifications of Tumors of the Central Nervous System. All grade 2 tumors were histologically classified as atypical meningiomas. The histology distribution is listed in Appendix A.

### 2.2. Data Collection

Clinical charts, surgical records, radiological findings, and pathological findings were reviewed to evaluate the following patient parameters: gender, age, preoperative symptoms, radiological features, the EOR (GTR or sub-total resection [STR]), time to tumor recurrence, and pathological diagnosis.

The meningioma location was classified as described previously [23]. Briefly, the sphenoid wing was divided into three roughly equal portions. The “deep, inner or clinoidal” inner third of the sphenoid wing was classified as medial, while the “middle or alar” and “outer or pterional” outer two-thirds were classified as lateral. The localization of each tumor was defined according to the position of the center of its attachment in the sphenoid bone. Bone invasion was defined as the presence of enhancing soft tissue within the sphenoid bone on a Magnetic Resonance Imaging (MRI) scan, similar in appearance to the soft tissue enhancement and intensity of intracranial tumors [24]. Hyperostosis was defined as sphenoid wing bone adjacent to the tumor that was thicker than normal cortical bone in Computed Tomography (CT) and MRI scans, with similar CT attenuation. Tumors with an irregular shape were defined as those with margin irregularity. Tumor volumetric analysis was performed using the volumetric function of Horos version 3.3.6 (http://www.horosproject.org accessed on 1 April 2022).

On the second day, the sixth month, and the first year after surgery, patients underwent contrast-enhanced magnetic resonance imaging (CE-MRI) for follow-up. If no tumor recurrence was observed, annual CE-MRI follow-ups were performed. All CE-MRIs were subjected to a central review. When the apparent enlargement of residual tumors was detected on CE-MRI scans, tumor recurrence was defined through inter-observer agreement between the neuro-radiologist and two neurosurgeons who were blind to the clinical or genetic data. Recurrence-free survival (RFS) was defined as the time between surgery and recurrence or final follow-up. Cases in which radiotherapy was administered before recurrence were considered censored at that time. The EOR was categorized as either GTR (Simpson grades I, II, and III) or STR (Simpson grade IV) based on postoperative imaging and surgical records [25].

A central review was conducted for all pathological diagnoses according to the 2021 WHO Classifications of Tumors of the Central Nervous System [26]. The MIB-1 labeling index (LI) was determined by using the highest labeling index values in the areas of maximum density, as identified through visual analysis [27]. Clinical and genetic information results were not yet available to the reference neuropathologists.

### 2.3. DNA Extraction and Sanger Sequencing

DNA was extracted from fresh-frozen or FFPE (formalin-fixed paraffin-embedded) tumors using a QIAamp DNA Mini Kit (QIAGEN; Venlo, The Netherlands) or a QIAamp DNA FFPE Tissue Kit according to the manufacturer’s instructions. DNA quality was evaluated using a spectrophotometer.

Based on previous reports [12,15,28,29,30], hotspot variants were evaluated in *AKT1* (NM_001382430, c.49G>A [p.Glu17Lys]), *KLF4* (NM_4235.6, c.1228A>C [p.Lys409Gln]), *SMO* (NM_005631, c.1234C>T [p.Leu412Phe] and c.1604G>T [p.Trp535Leu]), *POLR2A* (NM_000937.5, c.1207C>A [p.Gln403Lys], and c.1310–1315delACCTTC [p.Leu438_His439del]), *PIK3CA* (NM_006218, c.1624G>A [p.Glu542Lys], c.1633G>A [p.Glu545Lys/Ala], and c.3140A>G [p.His1047Arg]), and *TERTp* (NM_1014431, C228A(T) and C250T). Direct Sanger sequencing was performed for all exons in *NF2* (NM_000268.4) and for exons 12–21 (covering the WD40 domain) in *TRAF7* (NM_032271).

A total of 50 ng of DNA and KOD FX NEO (Applied Biosystems; Foster City, CA, USA) were used for polymerase chain reaction (PCR) in a 20 μL reaction mixture with the following reaction cycle: initial denaturation at 94 °C for 2 min, followed by 32 cycles of denaturation at 98 °C for 10 s, annealing at 58–60 °C for 30 s, and extension at 68 °C for 30 s, followed by final extension at 68 °C for 7 min (Appendix A). Sequences were determined using an ABI 3130xl Genetic Analyzer (Applied Biosystems). *NF2, AKT1, KLF4, SMO, POLR2A,* and *TERT* primer alignment is described in Appendix A. The PCR and primer alignment procedures for *PIK3CA* and *TRAF7* were performed as described previously [12,29,30].

### 2.4. Multiplex Ligation-Dependent Probe Amplification (MLPA)

The SALSA P044-C1 *NF2* probemix (MRC Holland; Amsterdam, The Netherlands) was used for MLPA analyses according to the manufacturer’s instructions. The P044-C1 NF2 probemix contained 21 probes for the *NF2* gene; five probes for chromosome 22q upstream of *NF2* (4 of which target *SMARCB1* and *LZTR1*); four probes for chromosome 22q downstream of *NF2*, and 13 reference probes for sequences on other chromosomes and some control fragments. DNA dosage was determined using Coffalyser.Net v210226.1433 (MRC Holland): 0.8–1.2 dosage quotients was considered normal, 0.4–0.7 dosage quotients indicated heterozygous or single exon deletions [31].

### 2.5. Statistical Analysis

All statistical analyses were performed using R v.4.1.1 (R Core Team, http://www.R-project.org accessed on 1 April 2022). Numerical variables were expressed as the median (interquartile range [IQR]) and compared using the Mann–Whitney U test. Categorical data were expressed as the count and percentage, and compared using Fisher’s exact test or the chi-square test. All reported *p* values were two-sided and values of < 0.05 were considered significant. Kaplan–Meier survival curves were plotted and differences in RFS between groups were compared using the log-rank test. Cutoff scores were determined according to the median score and clinical applicability. Multivariate analysis was performed using the Cox proportional hazard model.

## 3. Results

### 3.1. Tumor Characteristics of This Cohort

In the study cohort, genetic alterations were detected in *NF2* in 11 cases (23.4%), *AKT1* in 12 cases (25.5%), *TRAF7* in 11 cases (23.4%), *KLF4* in two cases (4.3%), *POLR2A* in one case (4.3%), *TERTp* in one case (4.3%), and as 22q loss in 15 cases (31.9%) (Figure 1). Details of the genetic information are provided in Appendix A. Among the 11 cases with *TRAF7* variants, nine (81.8%) also had *AKT1* variants. Among the cases with *NF2* alterations, variants were detected by Sanger sequencing in nine cases, and exon deletions were detected by MLPA in two cases. No variants were detected at *PIK3CA* or *SMO*. In 12 cases (25.5%), no gene variants could be identified, and these cases were designated as “not detected”.

Notably, all grade 2 and 3 meningiomas had *NF2* alterations and/or 22q loss (Figure 1). Among the four patients with grade 2 and 3 meningiomas, three experienced recurrence and one underwent irradiation therapy before recurrence and was considered censored at that time. WHO grade 1 and grade 2–3 meningiomas had the following characteristics: *NF2* alteration and/or 22q loss (grade 1, 32.6%; grade 2–3, 100%), bone invasion (grade 1, 9.3%; grade 2–3, 75%), irregular shape (grade 1, 20.9%; grade 2–3, 75%), MIB-1 LI > 4 (grade 1, 20.9%; grade 2–3, 50%), and GTR (grade 1, 62.8%; grade 2–3, 50%; Table 1).

### 3.2. Tumor Characteristics of WHO Grade 1 Meningiomas

A total of 43 patients in this study had WHO grade 1 sphenoid wing meningiomas, including 13 men and 30 women, with a median age of 57 (50–65.5) years, and a median follow-up months of 30 (14.5–70) (Table 2). Of the nine patients that experienced recurrence, seven had *NF2* alteration/22q loss, one had *AKT1* variation, and the genotype of the last case was not detected. In recurrent cases with *NF2* alteration/22q loss, GTR was achieved in three cases, and STR was achieved in four. When comparing the 14 patients with *NF2* alteration/22q loss and the 29 patients with wild-type *NF2*/22q, recurrence (*p* = 0.0026) and bone invasion (*p* = 0.008) differed significantly between the two groups.

### 3.3. Factors Related to RFS

Next, we performed a Kaplan–Meier analysis of RFS (Figure 2), finding that *NF2* alteration and/or 22q loss (*p* = 0.008) and MIB-1 LI > 4 (*p* = 0.03) were associated with a statistically significantly shorter time to recurrence, whereas gender and resection rate were not significantly associated with RFS (*p* = 0.07 and 0.3, respectively, Appendix A). In patients where GTR was achieved, *NF2* alteration and/or 22q loss were associated with a statistically significantly shorter time to recurrence (Figure 2C). Univariate Cox proportional hazards analysis (Table 3) revealed that *NF2* alteration/22q loss (hazard ratio [HR], 16.8; 95% confidence interval [CI], 2.1–137.0; *p* = 0.009) and MIB-1 LI > 4 (HR, 4.2; 95% CI, 1.05–16.9; *p* = 0.04) were significantly associated with tumor recurrence. Multivariate analysis performed using factors with *p* < 0.25 in the univariate analysis further indicated that *NF2* alteration/22q loss was associated with tumor recurrence (HR, 13.1; 95% CI, 1.5–111; *p* = 0.019).

## 4. Discussion

Sphenoid wing meningiomas have a higher rate of recurrence than meningiomas in other locations; however, the association between the genetic background of sphenoid wing meningiomas and their recurrence has not yet been fully elucidated. In this study, we found that WHO grade 1 sphenoid wing meningiomas with *NF2* alteration/22q loss are associated with a higher rate of recurrence than those with wild-type *NF2*/22q (*p* = 0.0026), even though the rate of GTR did not differ significantly between the two groups (*p* = 0.23). In addition, we found that the period of RFS was significantly shorter for patients with *NF2* alteration/22q loss (*p* = 0.008) and that *NF2* alteration/22q loss was significantly associated with recurrence (HR 13.1 [1.5–111]). Together, these results suggest that *NF2* alteration/22q loss is an independent factor that affects recurrence.

As for CNVs, it has been reported that cases with 22q loss have a HR of 12 (2–68, 95% CI) in recurrence and a complex karyotype HR of 32 (3–296, 95% CI) [32]. Moreover, cases with 22q loss but without *NF2* variants and complex karyotypes are reported to have a significantly poor prognosis for recurrence [33]. It may be worthwhile to evaluate CNVs other than 22q. Also, in recent years, several classifications based on integrated molecular analyses have been proposed and have indicated that the prognosis of patients with meningiomas with *NF2* variant/22q loss ranges from good to poor. In addition, these studies have suggested that a more accurate prognosis can be determined by analyzing CNVs, the methylome, and the transcriptome in addition to driver variants [17,18,34,35,36,37]. Nassiri et al. [18] divided meningiomas into four groups by combining DNA somatic CNVs, DNA somatic point mutations, DNA methylation, and messenger RNA abundance in a unified analysis. Based on this analysis, cases with *NF2* alteration/22q loss were classified into molecular groups (MGs) 1, 3, and 4, among which the prognosis for recurrence in MG3 and 4 meningiomas is reported to be poor. Sahm et al. [17] divided meningiomas into six groups mainly based on the methylome and classified WHO grade 1 meningiomas with *NF2* alteration/22q loss into methylation clusters (MCs) ben-1, ben-3, or int-A, among which int-A meningiomas have a relatively short RFS. The findings of these analyses suggest that the ability of *NF2* to predict recurrence also depends on CNVs, the methylome, and the transcriptome. Interestingly, our study suggested that, in sphenoid wing meningiomas, *NF2* alteration/22q loss can significantly predict recurrence; however, additional CNV, methylome, and transcriptome analysis could allow for more accurate prognostic analysis. Further research is required to elucidate the significance of driver variants in specific tumor locations.

Interestingly, we found that three patients with *NF2* alteration/22q loss (3/7: 42.9%) experienced recurrence even after GTR was achieved (Figure 1), suggesting that sphenoid wing meningiomas with *NF2* alteration/22q loss require close postoperative follow-up even if GTR is achieved (Figure 2F). Conversely, only one patient with *AKT1/TRAF7* alteration (1/14:7.1%, STR case) experienced recurrence, and no recurrence was observed among patients with *AKT1/TRAF7* alteration who achieved GTR (Figure 1). Therefore, conventional follow-up is considered necessary in patients with *AKT1/TRAF7* alteration that achieve STR, but the likelihood of recurrence is likely to be low in patients that achieve GTR. Meanwhile, no recurrence was observed in patients with *KLF4* and *POLR2A* variants. Taken together, these findings indicate that adjusting postoperative follow-up intensity according to genotype might be justified in patients with sphenoid wing meningiomas, and could improve the distribution of medical resources and provide medical and economic benefits. Genotyping meningiomas could also play a significant role in the consideration of different treatment strategies and in precision medicine; however, further studies are required to determine the modality and timing of treatments for recurrent meningiomas.

In this study, many of the meningiomas with *NF2* alteration/22q loss displayed bone invasion (Table 2), suggesting that meningiomas with bone invasion could harbor *NF2* variants preoperatively. Previously, Jin et al. reported that the radiographic presence of tumor invasion into the sphenoid wing and its association with underlying *NF2* variants can provide insights into the biological aggression potential of a tumor [24]. Therefore, future studies should investigate the molecular mechanisms that relate *NF2* variants with bone invasion. We found that oculomotor dysfunction was more common in patients with *NF2* alteration/22q loss, which may be confounded by bone invasion. The preoperative prediction of genotype from radiological features could therefore influence surgical strategy and tactics. However, even if it is presumed that the patient has *NF2* alteration/22q loss preoperatively, GTR should remain the main aim within safe limits. As such, future reports should aim to explore the association between genotype and clinical findings such as radiological features.

Our study had several limitations that should be addressed in future investigations. First, our study suffers from a small sample size and a retrospective, single-institution design. This small sample size could be the reason for the discovery of fewer cases with *PIK3CA* or *KLF4* variants. Therefore, the significance of these driver genes in the recurrence of WHO 1 sphenoid wing meningiomas could not be clarified in this study.

Second, our study did not analyze all driver genes, CNVs except 22q, the methylome, or the transcriptome. The driver genes evaluated in this study did not include all genes that had not been reported to be associated with meningiomas before. A more accurate prognostic analysis may be possible if the number of target genes is expanded. Especially in cases with *NF2* alteration/22q loss, adding CNVs, the methylome, or the transcriptome to that of this study may allow for more accurate prognostic analysis.

Third, since reference genetic material was not used in the genetic screening, it was not possible to confirm the somatic/germline nature of the variants identified, which was another limitation.

## 5. Conclusions

In this study, we found that in WHO grade 1 sphenoid wing meningiomas, *NF2* alteration/22q loss is an independent factor that is associated with recurrence and that patients with these meningiomas may require close postoperative follow-up even if GTR is achieved. Further studies are required to elucidate the significance of driver variants in specific tumor locations.

## Figures and Tables

**Figure 1 cancers-14-03183-f001:**
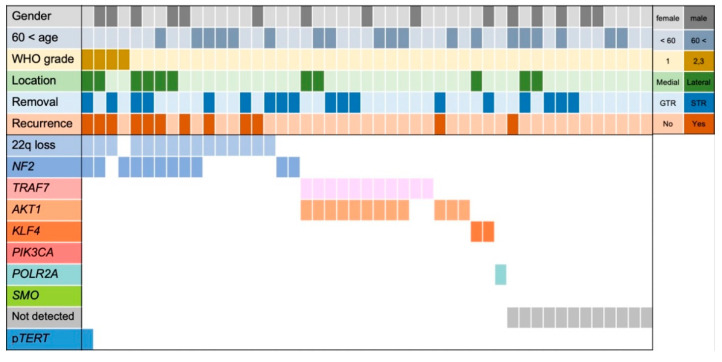
Clinical characteristics and genetic status of patients with sphenoid wing meningiomas. Besides one case, *NF2* alteration and/or 22q loss, *AKT1/TRAF7, KLF4,* and *POLR2A* were mutually exclusive. A total of 12 cases were defined as “not detected”. GTR: gross-total resection. STR: sub-total resection.

**Figure 2 cancers-14-03183-f002:**
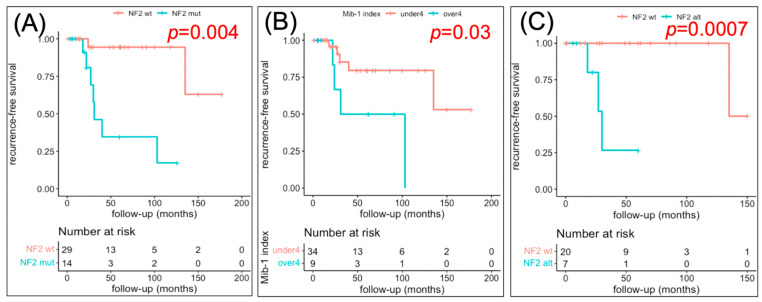
Kaplan–Meier plots of time to recurrence. (**A**) *NF2* alteration/22q loss vs. wild-type *NF2*/22q. RFS was significantly shorter in patients with *NF2* alteration/22q loss (*p* = 0.004). (**B**) MIB-1 LI > 4 vs. < 4. RFS was significantly shorter in patients with MIB-1 LI > 4 (*p* = 0.03). (**C**) *NF2* alteration/22q loss vs. wild-type *NF2*/22q in patients with GTR. RFS was significantly shorter in patients with *NF2* alteration/22q loss (*p* = 0.0007). GTR: gross-total resection.

**Table 1 cancers-14-03183-t001:** Patient characteristics.

Variables	WHO Grade 1–3 (47)	WHO Grade 1 (43)	WHO Grade 2/3 (4)
**General**			
Age (years)	57 (49–64)	57 (50–65.5)	50.5 (48.5–53)
Male	15 (31.9%)	13 (30.2%)	2 (50%))
Recurrence	12 (25.5%)	9 (20.9%)	3 (75%)
Follow-up months	27 (12–61.5)	28 (13–64)	16 (8.3–26.5)
**Clinical symptoms**			
Eye field symptom	22 (46.8%)	19 (44.2%)	3 (75%)
Vision loss	14 (29.8%)	12 (27.9%)	2 (50%)
Oculomotor dysfunction	4 (8.5%)	3 (7.0%)	1 (25%)
**Radiological features**			
Medial	36 (76.6%)	34 (79.1%)	2 (50%)
Bone invasion	7 (14.9%)	4 (9.3%)	3 (75%)
Hyperostosis	10 (21.3%)	8 (18.6%)	2 (50%)
Brain edema	17 (36.2%)	14 (32.6%)	3 (75%)
CS invasion	8 (17.0%)	6 (14.0%)	2 (50%)
Irregular shape	12 (25.5%)	9 (20.9%)	3 (75%)
Volume (cm^3^)	14.0 (6.0–33.1)	11.7 (5.8–28.5)	79.7 (42.8–125.6)
**Histological analysis**			
MIB-1 LI ≧ 4	11 (23.4%)	9 (20.9%)	2 (50%)
**Genetic analysis**			
*NF2 alt.*/22q loss	18 (38.3%)	14 (32.6%)	4 (100%)
Surgery			
GTR	29 (61.7%)	27 (62.8%)	2 (50%)

CS: cavernous sinus; LI: labeling index; GTR: gross-total resection.

**Table 2 cancers-14-03183-t002:** Characteristics of patients with WHO grade 1 meningiomas.

Variables	All (43)	*NF2* alt./22q Loss (14)	*NF2*/22q wt. (29)	*p* Value
**General**				
Age (years)	57 (50–65.5)	57 (47.8–63.8)	58 (51–67)	0.64
Male	13 (30.2%)	4 (28.6%))	9 (31.0%)	0.87
Recurrence	9 (20.9%)	7 (50%)	2 (6.9%)	**0.0026**
Follow-up months	30 (14.5–70)	25 (18.5–37.8)	30 (13–70)	0.30
**Clinical symptom**				
Eye field symptom	19 (44.2%)	7 (50.0%)	12 (41.4%)	0.59
Vision loss	12 (27.9%)	4 (28.6%)	8 (27.6%)	0.95
Oculomotor dysfunction	3 (7.0%)	3 (14.3%)	0 (0%)	**0.030**
**Radiological features**				
Medial	34 (79.1%)	10 (71.4%)	24 (82.8%)	0.39
Bone invasion	4 (9.3%)	4 (28.6%)	0 (0%)	**0.008**
Hyperostosis	8 (18.6%)	4 (28.6%)	4 (13.8%)	0.24
Brain edema	14 (32.6%)	3 (21.4%)	11 (37.9%)	0.28
CS invasion	6 (14.0%)	1 (7.1%)	5 (17.2%)	0.37
Irregular shape	9 (20.9%)	4 (28.6%)	5 (17.2%)	0.39
Volume (cm^3^)	11.7 (5.8–28.5)	10.2 (5.2–52.6)	11.7 (6.9–24.0)	0.21
**Histological analysis**				
MIB-1 LI ≧ 4	9 (20.9%)	4 (28.6%)	5 (17.2%)	0.39
**Surgery**				
GTR	27 (62.8%)	7 (50%)	20 (69.0%)	0.23

CS: cavernous sinus; LI: labeling index; GTR: gross-total resection.

**Table 3 cancers-14-03183-t003:** Multivariate analysis of tumor recurrence.

Variables	Univariate			Multivariate		
	HR	95% CI	*p* Value	HR	95% CI	*p* Value
*NF2*/22q loss	16.8	2.1–137.0	**0.009**	13.1	1.5–111	**0.019**
MIB-1 LI ≥ 4	4.2	1.05–16.9	**0.04**	2.2	0.5–9.3	0.28
Lateral	2.3	0.5–9.5	0.27			
Male	2.2	0.6–8.8	0.25			
Age < 60	2.3	0.6–9.5	0.23	1.6	0.4–6.6	0.53
STR	2.0	0.5–7.5	0.31			

HR: hazard risk; CI: confidence interval; LI: labeling index; STR: sub-total resection.

## Data Availability

Data are available on reasonable request. The authors confirm that the data supporting the findings of this study will be shared by request from any qualified investigator.

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
