# Peer review of "NF2 Alteration/22q Loss Is Associated with Recurrence in WHO Grade 1 Sphenoid Wing Meningiomas"

_cancers, 2022, doi:10.3390/cancers14133183_

Round 1

Reviewer 1 Report

The article “NF2 Alteration/22q Loss Is Associated with Recurrence in WHO 2 Grade 1 Sphenoid Wing Meningiomas” is focused on the relationship between the selected recurrent mutations in meningiomas and tumor recurrence.

The recurrence risk in low grade meningiomas is clinically important issue thus the goal of the study is justified. The study design and analysis are straightforward and simple.

The number of patients is small, especially number of grade 2 and 3 meningiomas however the study include the patients with tumors of very specific location only.

The authors determined the mutation status at hot spot regions of the selected genes involved in the pathogenesis of meningiomas as well as in entire coding sequence of NF2  and selected exons of TRAF7. Importantly both sequence and copy number analysis of NF2 were included.

The small number of patients and including only selected genetic loci are main weakness of the study, as already stated by the authors in the end of discussion section. Importantly the results are conclusive and statistical significance threshold was reached in the analysis.

The reference genetic material was not used in the mutational screening thus the somatic/germline nature of the identified variants was impossible which is another limitation.

My main concern relate to the reporting of the molecular results. I strongly suggest to include the information what exactly were the identified variants, especially those identified in NF2 and TRAF7 where a number of common SNPs could be identified. Please provide the positions according to HGMD nomenclature, rs id or cosmic id if available as well as information on the pathogenic or putative pathogenic status of the identified variants. Detail information can be presented as supplementary data.

If identified, nonpathogenic variants and  frequent SNP variants should be excluded from the analysis of clinical analysis. Did author looked at frequency of the identified variants in general populations and information on pathogenic/benign status of the variants?

Patients detail information on each patients characteristics could be included in supplementary materials be

Nonsignificant results should be excluded from figure 2, they can be presented as supplementary data.

Author Response

Response to Reviewer #1

The reference genetic material was not used in the mutational screening thus the somatic/germline nature of the identified variants was impossible which is another limitation.

Response:

Thank you for your comment to improve our manuscript. We added following sentence.

[Discussion: Page 8, Line 281–283]

Third, since reference genetic material was not used in the genetic screening, it was not possible to confirm the somatic/germline nature of the variants identified, which was another limitation.

My main concern relate to the reporting of the molecular results. I strongly suggest to include the information what exactly were the identified variants, especially those identified in NF2 and TRAF7 where a number of common SNPs could be identified. Please provide the positions according to HGMD nomenclature, rs id or cosmic id if available as well as information on the pathogenic or putative pathogenic status of the identified variants. Detail information can be presented as supplementary data.

If identified, nonpathogenic variants and  frequent SNP variants should be excluded from the analysis of clinical analysis. Did author looked at frequency of the identified variants in general populations and information on pathogenic/benign status of the variants?

Patients detail information on each patients characteristics could be included in supplementary materials be.

Response:

Thank you for the valuable comment. Details of the genetic information are provided in Supplementary Table 4.

Nonsignificant results should be excluded from figure 2, they can be presented as supplementary data.

Response:

Thank you for the valuable comment. We modified figure 2 appropriately, and added supplementary figure1.

Reviewer 2 Report

Thanks for the opportunitty to read this interesting study. My main concern is the small sample size, which makes it dfficult to make meaningful conclusins about most of the investigated genes. On the other hand, the authors are very clear in focusing on the main findings regarding NF2 variants in ther conclusions. This study serves as a good example of how genetics can be used in precision medicine.

There are a few minor language issues.

Author Response

Thank you for the review and comment.

Reviewer 3 Report

In this manuscript the authors utilize Sanger sequencing and multiplex  ligation-dependent probe amplification to analyze the –partial- molecular background of the worse-progmosis sphenoid wing meningiomas. The authors set out to use to characterize  47 samples into different classifications that associate with recurrence  and some clinical variable. They indeed found that NF2 alteration/22q loss are associated with a shorter RFS in WHO grade 1 sphenoid wing meningiomas tumors even in those cases were total resection was achieved.

Convincing data was presented to demonstrate that the samples could be subgrouped with significant p-values based on the genes/chromosomal regions found altered. While the tumors subgrouped this may be due to a small sample size. However this is a promising result which as the authors mention suggests a much larger cohort should be analyzed. The manuscript generates useful data for the scientific community but there are a number of issues that must be addressed before publication: 

1. Page 4 line 147-151, and 154-155: clinical and biological patient/sample description should be moved to the Materials and Methods section.

2. Page 4 line 147-15: the number of meningioma tumors do not agree as they add up to 48 instead of 47.

  3. Page 5 line 182-187: sentences are a bit confusing. It seems that the information is duplicated

  4. The Discussion section should be enriched by comparing the results of the authors with results from others (i.e: doi:10.3389/fonc.2021.740782; doi:10.1093/neuonc/not325).

Author Response

Response to Reviewer #3

Page 4 line 147-151, and 154-155: clinical and biological patient/sample description should be moved to the Materials and Methods section.

Response:

Thank you for the valuable comment. We moved following sentences to the Materials and Methods section.

[Materials and Methods, Page 2, Line 69 – 77]

This study included 47 patients with sphenoid wing meningiomas whose characteristics are listed in Table 1. The cohort included 15 men and 32 women with a median age of 57 (49–64) years and a median follow-up months of 27 (12–61.5). Of the 47 patients, 43 had WHO grade 1 meningiomas, 3 had WHO grade 2 meningiomas, 1 had a WHO grade 3 meningioma, and none had en plaque meningiomas. Although the grade 3 tumor was histologically classified as a meningothelial meningioma, it had a TERTp variant and was therefore classed as grade 3 according to the 2021 WHO Classifications of Tumors of the Central Nervous System. All grade 2 tumors were histologically classified as atypical meningiomas. The histology distribution is listed in Supplementary Table 3.

Page 4 line 147-15: the number of meningioma tumors do not agree as they add up to 48 instead of 47.

Response:

Thank you for the valuable comment. We collected the wrong number.

[Materials and Methods, Page 2, Line 71 – 73]

Of the 47 patients, 43 had WHO grade 1 meningiomas, 3 had WHO grade 2 meningiomas, 1 had a WHO grade 3 meningioma, and none had en plaque meningiomas.

 Page 5 line 182-187: sentences are a bit confusing. It seems that the information is duplicated

Response:

Thank you for the valuable comment. We deleted the following sentence.

[Results: Page 5, Line 181 - 184]

Nine of the patients with NF2/22q loss experienced recurrence (9/15: 46.7%), whereas one patient with AKT1/TRAF7 variation experienced recurrence (1/14: 7.1%).

  The Discussion section should be enriched by comparing the results of the authors with results from others (i.e: doi:10.3389/fonc.2021.740782; doi:10.1093/neuonc/not325).

Response:

Thank you for the valuable comment. We added the following sentences.

[Discussion: Page 7, Line 219 – 223]

As for CNVs, it has been reported that cases with 22q loss have a HR of 12 (2-68, 95% CI) in recurrence and complex karyotype HR of 32 (3-296, 95% CI).Also, cases with 22q loss but without NF2 variants and complex karyotypes are reported to have a significantly poor prognosis for recurrence. It may be worthwhile to evaluate CNVs other than 22q.